# Determinants of a mobile phone-based Interactive Voice Response (mIVR) system for monitoring childhood illnesses in a rural district of Ghana: Empirical evidence from the UTAUT model

**Timothy Kwabena Adjei**[1,2]*, **Aliyu Mohammed**[3], **Princess Ruhama Acheampong**[1], **Emmanuel Acquah-Gyan**[1], **Augustina Sylverken**[4,5], **Sampson Twumasi-Ankrah**[6], **Michael Owusu**[7], **Ellis Owusu-Dabo**[1]

1 Department of Global and International Health, School of Public Health- Kwame Nkrumah University of Science and Technology, Kumasi, Ghana, 2 Department of Obstetrics and Gynaecology, Komfo Anokye Teaching Hospital, Kumasi, Ghana, 3 Department of Epidemiology and Biostatistics, School of Public Health- Kwame Nkrumah University of Science and Technology, Kumasi, Ghana, 4 Department of Theoretical and Applied Biology, College of Science, Kwame Nkrumah University of Science and Technology, Kumasi, Ghana, 5 Kumasi Centre for Collaborative Research in Tropical Medicine, Kumasi, Ghana, 6 Department of Statistics and Actuarial Science, College of Science, Kwame Nkrumah University of Science and Technology, Kumasi, Ghana, 7 Faculty of Allied Health, Department of Medical Diagnostics, Kwame Nkrumah University of Science and Technology, Kumasi, Ghana

* timkadjei@gmail.com

**Data Availability Statement:** All files are available from the Inter-university Consortium for Political

## Abstract

### Background

The use of a mobile phone-based Interactive Voice Response (mIVR) System for real time monitoring of childhood illnesses provides an opportunity to improve childhood survival and health systems. However, little is known about the factors that facilitate its use. This study sought to identify key determinants and moderators of mIVR system use among caregivers in a rural district of Ghana using the Unified Theory of Acceptance and Use of Technology (UTAUT) model.

### Methods

The mIVR system was designed to provide real-time data on common symptoms of childhood illnesses after answering several questions by caregivers with sick children. A structured questionnaire with closed questions was used to collect data from 354 caregivers of children under-five living in rural communities, four (4) months after introducing the system. Regression analysis was used to identify key determinants and moderating factors that facilitate the use of the system based on the UTAUT model.

### Results

A total of 101 (28.5%) caregivers had used the system and 328 (92.7%) had intention to use the mIVR system. Caregivers' level of education and household wealth were associated

and Social Research (ICPSR) database (https://doi.org/10.3886/E124361V1).

**Funding:** Authors received no specific funding for this work. This work was self funded for submission as thesis as a student in the institution stated. International Development Research Centre (IDRC) however funded the main project (MOBCHILD) from which this study is nested. However they did not play any role in the design of the study.

**Competing interests:** The authors have declared that no competing interests exist.

with use of the mIVR systems (p<0.001). Behavioural intention (BI) to use mIVR system was positively influenced by performance expectancy (PE) (β = 0.278, 95% CI: 0.207, 0.349), effort expectancy (EE) (β = 0.242, 95% CI: 0.159, 0.326) and social influence (SI) (β = 0.081, 95% CI: 0.044, 0.120). Facilitating conditions (FC) (β = 0.609, 95% CI: 0.502, 0.715) and behavioural intention (β = 0.426, 95% CI: 0.255, 0.597) had a positive influence on user behaviour (UB). Mobile phone experience and household wealth significantly moderated the effect of PE, EE, SI, and FC on behavioural intention and usage of mIVR systems.

## Conclusion

The perceived usefulness of the mIVR system, ease of use, social influences, and facilitating conditions are key determinants of users' attitude and use of mIVR system. These relationships are significantly moderated by users' phone experience and wealth status.

## Background

Globally, it is estimated that about 85% of children less than 15 years who died in 2018 were under the age of 5, indicating that about 15,000 under-five deaths occur per day [1]. It is projected that children in sub-Saharan Africa are 15 times more likely to die before their fifth birthday than those in developed countries [1]. The situation is not different in Ghana, where a recent survey revealed high regional mortality rate (79 per 1,000 live births) despite gradual declining national rate of 56 per 1,000 live births. Infections such as acute respiratory infections (ARIs), malaria, and diarrhoeal diseases are amongst the main causes of under-5 morbidity and mortality [2]. Unfortunately, most of these childhood deaths occur in rural areas where there are limited health resources and facilities [3].

In the meantime, it is estimated by the World Health Organisation (WHO) that over 50% of childhood mortality could be prevented with affordable and simple interventions [1]. The use of mobile devices and technology to support medical practice or public health (mHealth) is one of such potential interventions in health care delivery that has attracted global attention because of the rapid rise in access to mobile phones [4]. Although there is an increase in the usage of mobile phone-based health information systems, its adoption rate is low [5]. The Interactive Voice Response (IVR) is one of the various forms of mHealth. While studies have revealed that the use of mHealth systems have the potential to improve the health system and disease outcomes [6], not much has been done in determining factors of mIVR use among its users [7].

Ghana has been recently recognized as one of the countries with high mobile phone subscription (34.57 million subscribers and a penetration rate of 119%), with nearly one third (10.1 million) of the entire population being active internet users [8]. This prospect necessitates the need to explore the acceptability of mIVR systems among end users. A growing number of studies have demonstrated the feasibility and potential of mIVR for improving health information systems [9,10] and its acceptability among users [11]. However, the key determinants of its use remain grey. Knowledge about key facilitators will enhance its uptake, scale-up and integration. A study has shown that mHealth interventions targeted at rural women, who most often are the primary caregivers of children, have the potential to reduce the barriers related to access to child healthcare services in rural settings [12].

In order to understand the multifactorial nature of how mIVR use is affected, the UTAUT model was adopted. This model, over the past decade helps to explain and predict the pathway which leads to adoption of technology [13–15]. Not many studies have applied the UTAUT model in relation to the IVR system, or disease surveillance especially in resource limited settings [15]. This study therefore sought to assess the determinants of a mobile phone-based IVR system use among caregivers of children under-five in the Asante Akim North District of Ghana.

## Materials and methods

### Study design and area

This was a cross-sectional study, nested in an on-going study (MOBCHILD project). The MOBCHILD project is a quasi-experimental study which seeks to improve childhood survival through the use of a mobile phone-based Health Information System (HIS). A large prospective cohort of caregivers of children under-five were given the opportunity to call a mobile phone-based IVR system to receive health advice for their sick under-five children via their mobile phones. The mIVR system interacts with the caregiver and the real time data obtained is intended to improve the health information system and also provide data for health authorities for planning. Caregivers are the main source of data for use. The study was conducted in the Asante Akim North District of the Ashanti Region of Ghana from 16th to 30th August, 2019. The district, whose inhabitants are predominantly farmers, has been profiled into 23 enumeration areas (EAs) with considerable network coverage. The study was conducted in seven randomly selected enumeration areas: Agogo Gyidim, Agogo Obuasi, Agogo Old Police Station, Bebuso, Domeabra, Zongo and Pataban/Aniwoso. Each EA had an average of four smaller communities.

### Study population

Study participants were selected from 1,026 caregivers involved in the MOBCHILD project within the 23 Enumeration Areas (EAs) in the Asante Akim North District. Caregivers were aged 18 years or above. Out of these eligible caregivers, 357 were randomly selected for this study. Caregivers were either legal parents or guardians of children aged 0–59 months. Caregivers who were non-residents of Asante Akim North and outside the boundaries of the enumeration areas were excluded. The study was conducted about 4 months after the introduction of the mIVR system to caregivers in the MOBCHILD project.

### Sample size and sampling

Using the cochran's formula $n_0 = (t^2pq)/d^2$ [16] with an estimated proportion (p) of 0.5, which represent 50% of the population of caregivers use the mobile phone-based IVR system, confidence level of 95% and precision (d) of 0.05, a sample size of 384 was attained.

In all, 357 caregivers constituting 34.8% of the sample frame for the MOBCHILD project were recruited for this study. A cluster sampling approach was adopted using the enumeration areas (EAs) as primary sampling units. To attain the required number of caregivers, 7 clusters were selected randomly out of the 23 EAs. All caregivers in the selected clusters were recruited as study participants.

### Data collection

A structured questionnaire was used to capture information on socio-demographic profiles of participants, mobile phone ownership and experience, socio-economic background, and

mIVR use. A section of the questionnaire was designed based on the UTAUT model. The UTAUT model is used to assess the degree to which performance expectancy, effort expectancy, social influence, and facilitating conditions affect behavioural intention and user behaviour to a technological system [14]. The UTAUT model posits that performance expectancy (PE), (the extent to which the individual believes that the use of the system will result in performance gains), effort expectancy (EE), (the degree of ease associated with the usage of a system), and social influence (SI) (the extent to which an individual believes that other people who are important to him or her, the caregiver, believes should use the new system) have direct positive influence on behavioural intention (BI), (an individual subjective possibility that he or she will exhibit the specific desired behaviour to use technology) [14]. Facilitating condition (FC) (the extent to which a person believes that there is the presence of an organisational and technical infrastructure assistance to use the technology or system) and behavioural intention (BI) directly influence user behaviour (UB), (the actual use behaviour of the system by an individual) [14]. These independent variables in the UTAUT model (PE, EE, FC and SI) were categorised as technology-oriented factors. A number of research works undertaken have elucidated various degrees to which these dependent variables (BI and UB) are influenced by their predictors [15]. In this study, a likert scale consisting of six constructs (PE, EE, SI, FC, BI, and UB) and 20 items were used to assess the degree of agreement or disagreement (1-Strongly disagree, 2-Disagree, 3-Neutral, 4-Agree, 5-Strongly Agree) in relation to the use of the mobile phone-based Interactive Voice Response (mIVR) system. The questionnaires were administered by trained research assistants with each lasting for 30 minutes.

## Data management

Data was captured using COMMCARE and transported to STATA 15 for data analysis. The socio-demographic and economic variables served as the moderating factors in this model: age, gender, education, socio-economic status and mobile phone experience. Age was categorised into ranges of 10, beginning from less than 20 years, 21–30 years, 31–40 years, 41–50 year, and 51 years and above. Gender was defined as either male or female. Highest educational level was categorised as no education, primary, Junior High School (JHS)/Middle School, Secondary, and Tertiary education. The principal component analysis (PCA) was used to estimate the socio-economic status (SES) or asset score of the respondents by using a wealth score centred on household assets. The ownership of about 24 household items was used to generate the asset score. Weighted scores were subsequently divided into 5 quintiles. The lowest quintile represents the poorest households, while the highest quintile denotes the wealthiest households. Mobile phone experience was assessed with 5 variables: phone usage (for calls, short messaging service (SMS) and internet purposes), duration of use of mobile phone and caregivers' self-reported knowledge of mobile phones. To generate experience with phone usage, the values for each variable were added to get a composite score. The score was re-categorised so that respondents could either be classified as inexperienced phone users or experienced mobile phone users with respect to the mean score. The mean score of the items for each construct in the UTAUT model was generated and used in the analysis to determine the predictors of mIVR use.

## Data analysis

Of the 357 responses received from the study participants, 3 were dropped after data cleaning, leaving 354 for analysis. The response rate was 99.2%. Descriptive statistics was used to describe the socio-demographic profile of the participants, where means and standard deviations were estimated for continuous variables and percentages for categorical variables. The

**Table 1. Results of cronbach alpha reliability.**

| Variable/Construct | Number of Items | Cronbach's Alpha (α) |
|---|---|---|
| Performance expectancy | 4 | 0.9132 |
| Effort expectancy | 4 | 0.8869 |
| Social influence | 3 | 0.5620 |
| Facilitating condition | 3 | 0.5860 |
| Behavioural intention | 3 | 0.5765 |
| Use behaviour | 3 | 0.8734 |

cronbach's alpha was calculated to assess the internal consistency of the various constructs in the UTAUT model as illustrated in Table 1 below. Linear regression analysis was conducted to assess the relationship between the independent (PE, EE, SI and FC) and dependent (BI and UB) variables in the UTAUT model. All pathways in the UTAUT model between the independent and dependent variables were tested for statistical significance. Moderating factors are variables that affect the strength or weakness of relationship between independent and dependent constructs in the model [17]. In other words, the relationship between the independent and dependent variables is affected by the presence of the moderators. The moderators (age, education, gender, socio-economic status, phone experience, ethnicity and religion) were introduced as confounders between these pathways. Moderators that resulted in significant change in the coefficient (β) of the association (R-squared ($R^2$) of more than 10%) were considered as significant moderators for those pathways.

## Ethical considerations

This study was approved by the Committee on Human Research, Publication and Ethics, Kwame Nkrumah University of Science and Technology (CHRPE-KNUST) with reference number CHRPE/AP/497/19 and by the authorities of the Health Directorate of Asante Akim North. All recruited participants who consented to the study were assured of confidentiality with respect to their identity and the data provided. Consent was given in a written form.

## Results

### Socio-demographic characteristics

The socio-demographic profile of participants is presented in Table 2. The mean age of the 354 caregivers was 30 years (SD±6.92) with majority being females. One-fifth of the caregivers had no formal education, and one-third of them had JHS or middle school education.

**Table 2. Socio-demographic profile of participants.**

| Variable | Frequency (N = 354) | Percentage (%) |
|---|---|---|
| **Age (years)** | | |
| ≤ 20 | 27 | 7.63 |
| 21–30 | 178 | 50.28 |
| 31–40 | 121 | 34.18 |
| 41–50 | 23 | 6.50 |
| > 50 | 5 | 1.41 |
| Mean (SD) | 30.17 (±6.92) | |
| **Gender** | | |
| Male | 48 | 13.56 |

(*Continued*)

**Table 2.** (Continued)

| Variable | Frequency (N = 354) | Percentage (%) |
|---|---|---|
| Female | 306 | 86.44 |
| **Highest Educational level** | | |
| No education | 77 | 21.75 |
| Primary | 87 | 24.58 |
| Middle/JHS | 120 | 33.90 |
| Secondary/Vocational | 57 | 16.10 |
| Tertiary | 13 | 3.67 |
| **Marital status** | | |
| Married/living together | 235 | 66.38 |
| Single | 35 | 9.89 |
| Divorced/separated | 83 | 23.45 |
| Widowed | 1 | 0.28 |
| **Religion** | | |
| Christianity | 259 | 73.16 |
| Muslims | 89 | 25.14 |
| Traditional | 6 | 1.70 |
| **Employment status** | | |
| Unemployed | 60 | 16.95 |
| Farmer | 156 | 44.07 |
| Hairdresser/seamstress | 19 | 5.37 |
| Apprentice/student | 12 | 3.39 |
| Trader | 97 | 27.40 |
| Civil servant | 10 | 2.82 |
| **Ethnicity** | | |
| Akan | 248 | 70.06 |
| Mole Dagbani | 78 | 22.03 |
| Others | 28 | 7.91 |
| **Cluster (Enumeration Area)** | | |
| Agogo Gyidim | 87 | 24.58 |
| Agogo Obuasi | 47 | 13.28 |
| Agogo old police station | 25 | 7.06 |
| Bebuso | 34 | 9.60 |
| Domeabra | 72 | 20.34 |
| Zongo | 56 | 15.82 |
| Pataban/Aniwoso | 33 | 9.32 |
| **Wealth quintiles (SES)** | | |
| 1st quintile(Poorest) | 71 | 20.06 |
| 2nd quintile | 71 | 20.06 |
| 3rd quintile | 83 | 23.45 |
| 4th quintile | 61 | 17.23 |
| 5th quintile (Wealthiest) | 68 | 19.21 |

SD, Standard deviation; SES, Socio-economic status.

## Access to mobile phone for use

While all participants had access to mobile phones, majority (86%) owned a phone and this was associated with increasing age of respondents (p<0.001), socio-economic status

(p<0.001) and male gender (p<0.050). Level of education (p>0.050) and phone experience (p>0.050) however did not have a significant association with phone ownership (S1 Table).

## Mobile phone experience

Majority of the participants used mobile phones for receiving and/or placing calls, and SMS. Almost half of the caregivers had used mobile phones for > 4 years. Overall, 57.34% of the caregivers were categorised as experienced mobile phone users, while 42.66% were found to be inexperienced.

## Usage of the IVR system by caregivers

Majority of the participants had never used (71.5%) the system before; however, most had the intention to use (92.7%) it in the future. The predominant reason for using the system was to seek for healthcare for their wards (S2 Table).

Table 3 shows the association between the demographic characteristics of the participants and mIVR usage. More than half of the IVR users were in the 21–30 year age group. Caregivers level of education and wealth quintiles of their households were associated with the use of the

**Table 3. Association between demographic characteristics and usage of the IVR system.**

| Variables | Usage of IVR system | | P- value |
|---|---|---|---|
| | Yes | No | |
| | n (%) | n (%) | |
| **Age (years)** | | | 0.640 |
| ≤ 20 | 6 (5.9) | 21 (8.3) | |
| 21–30 | 57 (56.4) | 121 (47.8) | |
| 31–40 | 32 (31.7) | 89 (35.2) | |
| 41–50 | 5 (5.0) | 18 (7.1) | |
| > 50 | 1(1.0) | 4 (1.6) | |
| **Educational level** | | | 0.001 |
| No education | 5 (5.0) | 72(28.5) | |
| Primary | 25 (24.8) | 62 (24.5) | |
| Middle/JHS | 44 (43.6) | 76 (30.0) | |
| Secondary/Vocational | 22 (21.8) | 35 (13.8) | |
| Tertiary | 5 (5.0) | 8 (3.2) | |
| **Wealth quintiles** | | | 0.001 |
| 1st quintile (Poorest) | 7 (6.9) | 64 (25.3) | |
| 2nd quintile | 24 (23.8) | 47 (18.6) | |
| 3rd quintile | 33 (32.7) | 50 (19.8) | |
| 4th quintile | 27 (26.7) | 34 (13.4) | |
| 5th quintile (Wealthiest) | 10 (9.9) | 58 (22.9) | |
| **Phone Experience** | | | 0.980 |
| Experience | 58 (57.4) | 145 (57.3) | |
| Inexperience | 43 (42.6) | 108 (42.7) | |
| **Gender** | | | 0.350 |
| Male | 11(10.9) | 37(14.6) | |
| Female | 90(89.1) | 216(85.4) | |

IVR, Interactive Voice Response.

**Table 4. Results of testing of the pathways in the model.**

| Path (Relationship) | Beta Coefficient (β) | Standard Error | 95%Confidence Interval | P-Value | Comments |
|---|---|---|---|---|---|
| PE→BI | 0.278 | 0.036 | 0.207–0.349 | 0.001 | Supported |
| EE→BI | 0.242 | 0.042 | 0.159–0.326 | 0.001 | Supported |
| SI→BI | 0.081 | 0.019 | 0.044–0.120 | 0.001 | Supported |
| FC→UB | 0.609 | 0.054 | 0.502–0.715 | 0.001 | Supported |
| BI→UB | 0.426 | 0.087 | 0.255–0.597 | 0.001 | Supported |

PE, Performance Expectancy; EE, Effort Expectancy; SI, Social Influence; FC, Facilitating Condition; BI, Behavioural Intention; UB, User Behaviour.

system (p = 0.001). There was no significant gender associations with usage of the IVR system (p = 0.350).

### Determinants of mIVR use

**Technology-oriented factors.** The constructs in the UTAUT model were used to assess the determinants of actual use of the mIVR system. The results as illustrated in Table 4 indicated that the relationships between PE and BI, EE and BI, SI and BI, FC and UB, and BI and UB were significant.

**Person-oriented factors (moderators).** The effect of the moderating factors for each of the pathways between the independent and dependent variables in the UTAUT model is shown in Table 5. Age and gender had no moderating effect on the relationship between PE, EE, SI and behavioural intention and usage. Education had a significant moderating effect between EE and BI. The relationships between PE and BI, EE and BI, and FC and UB were significantly moderated by phone experience and wealth status. Further, ethnicity and religion significantly moderated the association between FC and UB.

### Discussion

This study sought to identify key determinants and moderators of caregivers' use of a mobile phone-based interactive voice system (mIVR) using the UTAUT model.

The study revealed high phone ownership (86%) among caregivers in the area, consistent with a nationwide survey recently conducted in the country [2]. This finding indicates that mobile phone ownership in the study area has significantly increased from 42% as previously reported in a nationwide survey over the past decade [18]. Access to mobile phone remains an essential factor in the adoption and utilization of mHealth services. It was established that increase in age, male gender, and socio-economic status, as indicated by asset scores, were associated with mobile phone ownership (p<0.05), consistent with other studies [19].

Our results indicated that although males were likely to own mobile phones, usage of mIVR systems had no gender-based associations. However, caregivers' level of education and socio-economic status influenced their use of the mIVR system. Caregivers' use of the mHealth systems had no associations with age or phone experience. Our finding was however at variance with what Georgsson and Staggers [7] identified. A number of reasons could explain our findings. The ease of mobile phone operability, the type of mobile technology being employed (IVR), and the required task to be performed by the user, are likely reasons to influence utilization rates. The IVR application employed in the study required caregivers to make a call. Caregivers (91.8%) primarily reported using mobile phones for making or receiving calls. This is suggestive of their competence to use the system.

**Table 5. The effect of moderators on the association between the independent and dependent variables.**

| Moderator | Pathway (Relationship) | Crude Coefficient | Coefficient with Moderator | Comments on Moderation |
|---|---|---|---|---|
| Age | PE→BI | 0.195 | 0.198 | Not supported |
| | EE→BI | 0.187 | 0.187 | Not supported |
| | SI→BI | 0.082 | 0.085 | Not supported |
| | FC→UB | 0.853 | 0.842 | Not supported |
| Education | PE→BI | 0.195 | 0.183 | Not supported |
| | EE→BI | 0.187 | 0.164 | **Supported** |
| | SI→BI | 0.082 | 0.087 | Not supported |
| | FC→UB | 0.853 | 0.786 | Not supported |
| Phone Experience | PE→BI | 0.195 | 0.257 | **Supported** |
| | EE→BI | 0.187 | 0.252 | **Supported** |
| | SI→BI | 0.082 | 0.082 | Not supported |
| | FC→UB | 0.853 | 0.700 | **Supported** |
| Wealth quintile(SES) | PE→BI | 0.195 | 0. 240 | **Supported** |
| | EE→BI | 0.187 | 0.214 | **Supported** |
| | SI→BI | 0.082 | 0.085 | Not supported |
| | FC→UB | 0.853 | 0.758 | **Supported** |
| Gender | PE→BI | 0.195 | 0.193 | Not supported |
| | EE→BI | 0.187 | 0.176 | Not supported |
| | SI→BI | 0.082 | 0.089 | Not supported |
| | FC→UB | 0.853 | 0.795 | Not supported |
| Ethnicity | PE→BI | 0.195 | 0.207 | Not supported |
| | EE→BI | 0.187 | 0.184 | Not supported |
| | SI→BI | 0.082 | 0.088 | Not supported |
| | FC→UB | 0.853 | 0.735 | **Supported** |
| Religion | PE→BI | 0.195 | 0. 213 | Not supported |
| | EE→BI | 0.187 | 0.191 | Not supported |
| | SI→BI | 0.082 | 0.089 | Not supported |
| | FC→UB | 0.853 | 0.707 | **Supported** |

PE, Performance Expectancy; EE, Effort Expectancy; SI, Social Influence; FC, Facilitating Condition; BI, Behavioural Intention; UB, User Behaviour; SES, Socio-economic Status.

*Moderation was considered supported if coefficient (ß) change was more than 10%.

The findings also suggest that, although only a third had reported ever using the service mainly for the purpose of seeking health care services for the sick children, most of the caregivers expressed intentions to use mHealth service in the future (92.7%). These findings were not surprising because caregivers were interviewed less than four after introduction to the mIVR system. Caregivers who had not used the system affirmed this in our findings (64% of non-users), indicating that their children had not fallen sick for them to use the system (S1 Fig).

The results of the study suggest performance expectancy, effort expectancy, and social influence significantly influenced caregivers' intention to use mIVR systems. It is therefore important to note that the usefulness of the mIVR system in addressing the needs of the user as well as the ease associated with the use of the system will significantly contribute to users' behavioural intention and actual use of such services. The usefulness of this mIVR system by implication goes beyond the caregiver by providing an avenue to generate health data for health system planning. To achieve user satisfaction, developers of mobile technology systems must consider the needs of the target population and such systems must also be user-friendly to

ensure acceptability [7]. Mobile phone-based IVR systems must therefore be easy to use, requiring less education especially in rural settings to ensure acceptability and usability. In addition, facilitating conditions and users' behavioural intention were also found to positively impact actual use behaviour among caregivers. In other words, having the requisite knowledge and resources, users' perceptions of existing organization or system that facilitate their use of mHealth services, and behavioural intention are strong predictors of actual use behaviour. Numerous studies that applied the UTAUT model to ascertain the key factors influencing the acceptance and use of mHealth also established similar findings [15,20,21]. Even though Alshehri *et al*., [22] reported similar findings that PE and EE had a positive impact on behavioural intention, SI however did not significantly influence behavioural intention. Due to variations in social and health systems across different populations, such outcomes may be anticipated.

Our findings revealed that gender and age had no moderating effect on the relationship among performance expectancy, effort expectancy, social influence, and facilitating conditions in behavioural intention and usage of mHealth service, consistent with the findings of Alshehri *et al*., [22]. Alam *et al*., [23] however identified that gender significantly moderated the effect of PE, EE and FC on users' behavioural intention to adopt mHealth in Bangladesh. These findings are therefore suggestive that age and gender disparities do not weaken or strengthen the effects of key determinants of mHealth adoption and use among users in rural Ghana.

This study discovered that socio-economic status and phone experience significantly moderated the effect of performance expectancy and effort expectancy on behavioural intention to use mHealth. The association between performance expectancy and effort expectancy on behavioural intention to use mHealth service was stronger among users of high socio-economic status and phone experience than those of low socio-economic status and inexperience in mobile phones. The connection between facilitating condition and user behaviour was also significantly controlled by socio-economic status and phone experience. This outcome was consistent with a study found that experience significantly moderated the effect of FC on behavioural intention [22]. Education only significantly moderated the effect of effort expectancy on behavioural intention to use mIVR system, indicating that the relationship between effort expectancy and behavioural intention to use mIVR system is significant and stronger for highly educated users than for less educated ones.

In addition to the factors itemized for moderation, ethnicity and religion were identified to significantly moderate the effect of facilitating condition on actual mIVR user behaviour, suggesting that one's religious association or ethnic background can significantly impact the relationship between FC and user behaviour in mHealth services. It is worth recognizing that such socio-cultural factors can significantly mediate the effects of key determinants of mHealth adoption and use. Religious and ethnic consideration may therefore play major roles in facilitating mHealth adoption especially in settings similar to the study area. However, this may not be the case in other jurisdictions, or user populations because of cultural diversities.

## Limitations

This study had some limitations. First, the self-reported use of the mIVR system by caregivers was not validated and therefore might not accurately reflect the actual use. The study also identified barriers such as unstable mobile network in some communities which could pose challenges for caregivers in using the mIVR system. This barrier however was not widespread. As with all cross-sectional studies, temporality cannot be assured and thus the result of this study should be interpreted with caution. Despite the acknowledged limitations, rigorous methodology was applied in accordance with best practice for the conduct of epidemiological research.

The findings therefore will be beneficial to the scientific knowledge community and stakeholders.

## Conclusions

This study established that educational level and socio-economic status are significantly associated with the use of mobile phone-based IVR systems. Further, the study revealed that caregivers of children under five years have high intentions to use mIVR systems in the future. The study also confirmed that perceived usefulness of mIVR systems, the ease associated with its use, social influence and facilitating conditions are strong determinants of users' attitude and actual use of mIVR systems. The study also established caregivers' mobile phone experience and socio-economic backgrounds have significant moderating effect on the key determinants to the use of mIVR systems.

This study recommends that stakeholders should harness the usefulness of incorporating mIVR systems into the health information system to generate real time health data and inform planning and decision making processes within the health system. Developers of mHealth systems may also consider these user characteristics for mHealth integration and scaling up especially in limited resource settings.

## Supporting information

**S1 Fig. Caregivers' reasons for non-use of mIVR system.**
(DOCX)

**S1 Table. Analysis of mobile phone ownership among caregivers.**
(DOCX)

**S2 Table. Distribution of use of the IVR system by caregivers.**
(DOCX)

## Acknowledgments

We would like to acknowledge ESOKO Ghana, for the development of the mIVR system. We are also grateful to the Health Directorate of Asante Akim North District for their support. Our special gratitude also goes to Patricia Amoah Yirenkyi and Eunice Darkowaa for their immense administrative assistance.

## Author Contributions

**Conceptualization:** Timothy Kwabena Adjei, Princess Ruhama Acheampong, Emmanuel Acquah-Gyan, Michael Owusu, Ellis Owusu-Dabo.

**Data curation:** Timothy Kwabena Adjei, Sampson Twumasi-Ankrah.

**Formal analysis:** Timothy Kwabena Adjei, Aliyu Mohammed, Princess Ruhama Acheampong, Sampson Twumasi-Ankrah.

**Funding acquisition:** Timothy Kwabena Adjei.

**Investigation:** Emmanuel Acquah-Gyan, Michael Owusu.

**Methodology:** Timothy Kwabena Adjei, Aliyu Mohammed, Princess Ruhama Acheampong, Sampson Twumasi-Ankrah, Michael Owusu.

**Project administration:** Emmanuel Acquah-Gyan, Augustina Sylverken.

**Resources:** Princess Ruhama Acheampong, Emmanuel Acquah-Gyan, Ellis Owusu-Dabo.

**Supervision:** Princess Ruhama Acheampong, Augustina Sylverken, Sampson Twumasi-Ankrah, Michael Owusu, Ellis Owusu-Dabo.

**Validation:** Princess Ruhama Acheampong, Augustina Sylverken, Ellis Owusu-Dabo.

**Visualization:** Augustina Sylverken.

**Writing – original draft:** Timothy Kwabena Adjei.

**Writing – review & editing:** Timothy Kwabena Adjei, Aliyu Mohammed, Princess Ruhama Acheampong, Augustina Sylverken, Sampson Twumasi-Ankrah, Michael Owusu, Ellis Owusu-Dabo.

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
