## [Decision Letter · Decision Letter 0]

16 Dec 2020

PONE-D-20-32440

Determinants of a Mobile phone-based Interactive Voice Response (mIVR) system for monitoring childhood illnesses in a rural district of Ghana: empirical evidence from the UTAUT model

PLOS ONE

Dear Dr. Adjei,

Thank you for submitting your manuscript to PLOS ONE. After careful consideration, we feel that it has merit but does not fully meet PLOS ONE’s publication criteria as it currently stands. Therefore, we invite you to submit a revised version of the manuscript that addresses the points raised during the review process.

We look forward to receiving your revised manuscript.

Kind regards,

Olalekan Uthman, MD, MPH, PhD, FRSPH, FHEA

Academic Editor

PLOS ONE

Journal Requirements:

2) Please include additional information regarding the survey or questionnaire used in the study and ensure that you have provided sufficient details that others could replicate the analyses. For instance, if you developed a questionnaire as part of this study and it is not under a copyright more restrictive than CC-BY, please include a copy, in both the original language and English, as Supporting Information.

Reviewers' comments:

Reviewer's Responses to Questions

**Comments to the Author**

1. Is the manuscript technically sound, and do the data support the conclusions?

Reviewer #1: Yes

2. Has the statistical analysis been performed appropriately and rigorously? 

Reviewer #1: Yes

3. Have the authors made all data underlying the findings in their manuscript fully available?

Reviewer #1: No

4. Is the manuscript presented in an intelligible fashion and written in standard English?

Reviewer #1: Yes

5. Review Comments to the Author

Reviewer #1: The paper aims to identify the main determinants of a mobile phone-based Interactive Voice Response (mIVR) System using the Unified Theory of Acceptance and Use of Technology (UTAUT) model, in a rural area in Ghana. The results from this work show that performance and effort expectancies, respectively and social influence played a major role in influencing caregiver’s intention to use mIVR system. Also, ethnicity and religion had a significant effect on the relationship between facilitating conditions and user behavior in mHealth services.

Reviewer comments

The paper is clearly written. Below are a few comments:

In the study design area section, it is stated that the study was carried out in 7 randomly selected communities out of 115 communities: (a) I think the authors should state why they chose 7 out of 115 and not more.(b) The authors should also show the number of participants in each of the 7 communities in which they conducted their study.

I think this paper should be accepted (after the comments above have been addressed)

6. PLOS authors have the option to publish the peer review history of their article (what does this mean?). If published, this will include your full peer review and any attached files.

Reviewer #1: No

<gdiv></gdiv>

---

## [Author Response · Author response to Decision Letter 0]

26 Jan 2021

Question 3: Have the authors made all data underlying the findings in their manuscript fully available?

Reviewer #1: No

Response: The data underlying the findings in the manuscript have been deposited in public repository. This is accessible through this URL link (https://doi.org/10.3886/E124361V1) which was provided in the submission of the manuscript. All files are available from the Inter-university Consortium for Political and Social Research (ICPSR) database.

Reviewer’s Comment: In the study design area section, it is stated that the study was carried out in 7 randomly selected communities out of 115 communities: (a) I think the authors should state why they chose 7 out of 115 and not more. (b) The authors should also show the number of participants in each of the 7 communities in which they conducted their study.

Response: 

(a) The study area, Asante Akim North District has been profiled into 23 Enumeration Areas (EAs) by the district. Out of these, 7 were selected for this study. The 7 EAs were labeled as communities instead of enumeration areas in the manuscript. Each EA has an average of 5 smaller communities. This change has therefore been effected in the main manuscript. The selection of 7 EAs was also because certain areas had network challenges. 

(b) The various number of study participants in each enumeration area (EA) is illustrated in Table 2: Socio-demographic profile of participants.

Thank you very much.

---

## [Decision Letter · Decision Letter 1]

25 Feb 2021

Determinants of a Mobile phone-based Interactive Voice Response (mIVR) system for monitoring childhood illnesses in a rural district of Ghana: empirical evidence from the UTAUT model

PONE-D-20-32440R1

Dear Dr. Adjei,

We’re pleased to inform you that your manuscript has been judged scientifically suitable for publication and will be formally accepted for publication once it meets all outstanding technical requirements.

Kind regards,

Olalekan Uthman, MD, MPH, PhD, FRSPH, FHEA

Academic Editor

PLOS ONE

Reviewers' comments:

Reviewer's Responses to Questions

**Comments to the Author**

1. If the authors have adequately addressed your comments raised in a previous round of review and you feel that this manuscript is now acceptable for publication, you may indicate that here to bypass the “Comments to the Author” section, enter your conflict of interest statement in the “Confidential to Editor” section, and submit your "Accept" recommendation.

Reviewer #1: All comments have been addressed

2. Is the manuscript technically sound, and do the data support the conclusions?

Reviewer #1: Yes

3. Has the statistical analysis been performed appropriately and rigorously? 

Reviewer #1: Yes

4. Have the authors made all data underlying the findings in their manuscript fully available?

Reviewer #1: (No Response)

5. Is the manuscript presented in an intelligible fashion and written in standard English?

Reviewer #1: (No Response)

6. Review Comments to the Author

Reviewer #1: (No Response)

7. PLOS authors have the option to publish the peer review history of their article (what does this mean?). If published, this will include your full peer review and any attached files.

Reviewer #1: No

---

## [Editor Report · Acceptance letter]

3 Mar 2021

PONE-D-20-32440R1 

Determinants of a Mobile phone-based Interactive Voice Response (mIVR) system for monitoring childhood illnesses in a rural district of Ghana: empirical evidence from the UTAUT model 

Dear Dr. Adjei:

I'm pleased to inform you that your manuscript has been deemed suitable for publication in PLOS ONE. Congratulations! Your manuscript is now with our production department. 

Kind regards, 

on behalf of

Dr. Olalekan Uthman 

Academic Editor

PLOS ONE